

# Microbial communities in sediment from *Zostera marina* patches, but not the *Z. marina* leaf or root microbiomes, vary in relation to distance from patch edge

Cassandra L. Ettinger[1], Sofie E. Voerman[2,3], Jenna M. Lang[1,4], John J. Stachowicz[5] and Jonathan A. Eisen[1,5,6]

[1] Genome Center, University of California, Davis, CA, United States
[2] Climate Change Cluster, University of Technology Sydney, Sydney, Australia
[3] School of Life Sciences, University of Technology Sydney, Sydney, Australia
[4] Trace Genomics, San Francisco, CA, United States
[5] Department of Evolution and Ecology, University of California, Davis, CA, United States
[6] Department of Medical Microbiology and Immunology, University of California, Davis, Davis, CA, United States

Corresponding author
Jonathan A. Eisen,
jaeisen@ucdavis.edu,
jonathan.eisen@gmail.com

## ABSTRACT

**Background**. *Zostera marina* (also known as eelgrass) is a foundation species in coastal and marine ecosystems worldwide and is a model for studies of seagrasses (a paraphyletic group in the order *Alismatales*) that include all the known fully submerged marine angiosperms. In recent years, there has been a growing appreciation of the potential importance of the microbial communities (i.e., microbiomes) associated with various plant species. Here we report a study of variation in *Z. marina* microbiomes from a field site in Bodega Bay, CA.

**Methods**. We characterized and then compared the microbial communities of root, leaf and sediment samples (using 16S ribosomal RNA gene PCR and sequencing) and associated environmental parameters from the inside, edge and outside of a single subtidal *Z. marina* patch. Multiple comparative approaches were used to examine associations between microbiome features (e.g., diversity, taxonomic composition) and environmental parameters and to compare sample types and sites.

**Results**. Microbial communities differed significantly between sample types (root, leaf and sediment) and in sediments from different sites (inside, edge, outside). Carbon:Nitrogen ratio and eelgrass density were both significantly correlated to sediment community composition. Enrichment of certain taxonomic groups in each sample type was detected and analyzed in regard to possible functional implications (especially regarding sulfur metabolism).

**Discussion**. Our results are mostly consistent with prior work on seagrass associated microbiomes with a few differences and additional findings. From a functional point of view, the most significant finding is that many of the taxa that differ significantly between sample types and sites are closely related to ones commonly associated with various aspects of sulfur and nitrogen metabolism. Though not a traditional model organism, we believe that *Z. marina* can become a model for studies of marine plant-microbiome interactions.

# INTRODUCTION

The seagrass, *Zostera marina,* is a foundation species in protected bays and estuaries throughout the temperate northern hemisphere. Seagrasses are fully submerged marine angiosperms and are a paraphyletic group comprised of three lineages in the order *Alismatales* that convergently adapted to the marine environment between 70 and 100 million years ago (*Les, Cleland & Waycott, 1997*). There are only approximately 60 species of seagrass compared to the 250,000 species of terrestrial angiosperms, a testament to the strict selective pressure posed by re-entry to the marine environment (*Orth et al., 2006*). Seagrass patches serve as habitat and nursery grounds for many marine species, play key roles in nutrient cycling and carbon sequestration, and serve to protect the coastline from erosion (*Williams & Heck Jr, 2001*). *Z. marina* populations, like those of many seagrass species, are negatively affected by climate change, pollution and habitat destruction and so far, restoration efforts have been costly and ineffective (*Orth et al., 2006*). As a result, *Z. marina* is vulnerable to habitat fragmentation and loss.

The work described here was originally focused on a phenomenon known as "edge effects" in which the border between habitats is intermediate in abiotic conditions from the center of either adjacent habitat and thus the biotic composition of the border habitat, or edge, may differ from that of interior, intact habitat. Edges often support a mixture of organisms from two adjacent habitats (*Fox et al., 1997*; *Davies-Colley, Payne & Elswijk, 2000*), but may be abiotically unsuitable for species found in the center of either habitat. Increased predation and invasion by non-native species can also be a common feature of edges (*Paton, 1994*; *Fox et al., 1997*; *Harrison & Bruna, 1999*; *Flaspohler, Temple & Rosenfield, 2001*). Prior work on seagrasses have shown edge effects on species abundances (*Smith et al., 2008*; *Smith et al., 2011*; *Tanner, 2005*) and abiotic conditions such as turbulence (*Granata et al., 2001*; *Folkard, 2005*), carbon stocks (*Ricart et al., 2015*) and organic matter deposition (*Duarte & Sand-Jensen, 1990*; *Cebrián et al., 2000*). Here we investigate whether such edge effects are evident in the microbiota found in, on and near *Z. marina* plants.

Our interest in the microbiota for this study is driven by our overarching goal of developing *Z. marina* as a model for studies of microbial communities associated with marine plants. Although we speculate that plant microbe interactions are important for seagrasses, little is known about how the roles of microbial communities associated with marine plants may affect plant health and what abiotic and biotic factors affect the composition of these communities. Terrestrial plants, like *Arabidopsis* (*Lundberg et al., 2012*), corn (*Peiffer et al., 2013*; *Bouffaud et al., 2014*), rice (*Peiffer et al., 2013*; *Edwards et al., 2015*) and poplar (*Beckers et al., 2016*) have been shown to have distinct microbial communities on the inside (endophytes) and on the surface (epiphytes) of plant leaves and roots, as well as in the surrounding soil or sediment (rhizosphere) (*Lundberg et al., 2012*).

These communities can vary across different stages of plant development (*Chaparro, Badri & Vivanco, 2014*) and with local environmental conditions. In terrestrial systems the main drivers of plant associated microbial community composition are considered to be environmental factors, like soil particle size, pH and moisture content, as well as host plant species (*Aleklett et al., 2015*; *Lakshmanan, 2015*). Thus, examining eelgrass microbiota across a known environmental gradient from the center to the outside of a patch has the potential to provide insights into factors that shape the eelgrass microbiome, the full community of microorganisms associated with eelgrass. Recently a few culture-independent surveys of seagrass microbiomes have been published and these provide good initial reference points for our work here (*Jiang et al., 2015*; *Sun et al., 2015*; *Cúcio et al., 2016*; *Mejia et al., 2016*). Although, these studies have similar big picture findings, there are small differences in microbiome composition between them and thus further teasing apart of the factors that shape seagrass microbiomes is necessary and important work.

In this study, we characterized and then compared the epiphytic and rhizospheric bacterial communities of eelgrass using root, leaf and sediment samples obtained from the inside, edge and outside of a single subtidal *Z. marina* patch. We focused on characterizing the bacterial and archaeal members of the microbiome in each of these samples using high throughput sequencing of 16S ribosomal gene PCR libraries. We focus in particular on the following questions: What is the general taxonomic composition of the *Z. marina* microbiome? Are there changes in sediment microbial community composition or in biodiversity at the patch edge and, if so, what factors are driving observed differences, environmental abiotic factors or presence/absence of *Z. marina*? This analysis reveals multiple novel insights into the general structure of the *Z. marina* microbiome and lays the groundwork for further studies.

## MATERIALS & METHODS

### Sample collection

We collected leaf, root and sediment samples for microbiome analysis from 0.25 m$^2$ quadrats ($n = 4$) located in the interior (2.5 m from the edge), on the edge (but within the eelgrass habitat) and outside (2.5 m from the edge) of a single shallow subtidal eelgrass patch in Bodega Bay, CA (GPS: 38.319435, −123.053838) during the summer of 2013. Quadrats were positioned 2.5 m from each other parallel to the patch' edge. Samples were collected during low tide (±0.5 m water depth) at night (11 PM). For quadrats located at the center or edge of the eelgrass patch, one eelgrass shoot was sampled and directly separated into root and shoot tissue. The root tissue consisted of one entire root bundle sampled, the leaf tissue consisted of a clipped leaf of ± 3 cm in length positioned at about half way along the shoot length (±20 cm from the base). For each quadrat, sediment samples were collected at two sediment depths, 0.5 cm or less and 3 cm deep, from randomly selected locations within the quadrats. Microbial samples were directly stored on ice and transported to the laboratory within one hour where samples were frozen at −20 °C until further analysis.

Environmental data and the samples used for microbiome analysis were collected simultaneously. For each quadrat, eelgrass density was estimated by direct count.

Temperature, pH, salinity and dissolved oxygen were measured at 20 cm above the sediment with a YSI 556 handheld multimeter (YSI Inc., Yellow Springs, OH, USA), at a similar height as the shoot tissue was sampled. Sediment chemical and physical properties were assessed by separately coring the top 4 cm of sediment (10 cm diameter, taken twice within a quadrat and combined for analysis), to correspond with the sediment layer most influenced by the eelgrass roots. Sediment was dried (three days at 40 °C), mixed, sieved (sieve sizes: 710, 500, 355, 250, 180, 90 and 30 μm) and particle size fractions were weighed to investigate particle size distribution. A portion of the mixed sediment samples (±50 g) was separately analyzed for total organic carbon (TOC), total inorganic carbon (TIC) and Carbon:Nitrogen (C:N) ratio by the UC Davis Analytical Laboratory.

### Molecular methods

DNA was extracted from leaf ($n = 8$), root ($n = 8$) and both shallow ($n = 12$) and deep sediment ($n = 12$) samples as well as from a kit control ($n = 1$) with the PowerSoil DNA Isolation kit (MO BIO Laboratories, Inc., Carlsbad, CA, USA) according to the manufacturer's protocol. For the DNA extractions, root and leaf tissues were placed directly into PowerBead tubes from the freezer without grinding. Microbial 16S rRNA genes were amplified using a two-step protocol targeting the V4 region using the "universal" 515F and 806R primers (*Caporaso et al., 2012*). The primer set was modified to include Illumina adapters and barcode sequences using a dual indexing approach as in *Lang, Eisen & Zivkovic (2014)*. The 41 samples described in this paper were multiplexed with 103 samples from other experiments, for a total of 144 samples on the run. Libraries were sequenced by the UC Davis Genome Center Core Facilities on an Illumina MiSeq (Illumina, Inc., San Diego, CA, USA) to generate 250 bp paired-end reads.

### Sequence processing

A custom in-house script was used to demultiplex, quality check and merge paired reads (https://github.com/gjospin/scripts/blob/master/Demul_trim_prep.pl). The resulting sequences were analyzed using the Quantitative Insights Into Microbial Ecology (QIIME) v. 1.9.0 workflow (*Caporaso et al., 2010*).

For a detailed walkthrough of the following analysis using QIIME, see the IPython notebook (http://nbviewer.jupyter.org/gist/casett/86da7fc8749d27574f183498df65134a).

The sequencing run for this project included samples from other projects. In total, for the entire run, 14,163,470 reads passed quality filtering (Q20). Of these reads, 4,573,318 were associated with the 41 samples for this project. Of the 4,573,318 reads for this project, 4,212,549 merged successfully (92.11%). The sample with the lowest number of sequences after merging was the negative control with 444 sequences, the next lowest sample, BB039, had 22,897 sequences, approximately a fifty-fold increase. The most abundant sequence in the negative control was chloroplast DNA, and thus, we conclude that these 444 sequences were likely the result of contamination from other samples during sequencing or molecular analyses. We considered removing shared operational taxonomic units (OTUs) or 100 percent identical DNA sequences between the negative control and our environmental samples, but determined both of these actions to be too stringent on the dataset when

taking into account the abundance of the OTU's/DNA sequences in the samples and the number of reads in the negative control. Instead the negative control was simply removed from downstream analysis.

A total of 4,976 chimeras were identified using USEARCH v. 6.1 and were filtered out. The remaining sequences were clustered using the open reference approach into OTUs at 97 percent similarity using UCLUST (*Edgar, 2010*). Taxonomy was assigned using the assign_taxonomy.py QIIME script with the GreenGenes database (v.13_8) (*DeSantis et al., 2006*) using UCLUST. Further filtering was performed using the QIIME scripts, filter_taxa_from_otu_table.py and filter_otus_from_otu_table.py, to remove chloroplast DNA, mitochondrial DNA and singletons. Reads classified as "Unassigned" at the domain level were also removed from downstream analysis. After these filtering steps, the lowest number of sequences in a sample dropped to 3,277. This reduction in the number of sequence reads can be largely attributed to the removal of *Z. marina* chloroplast DNA from the leaf and root samples.

To aid in statistical comparison between different sample types (leaf, root, sediment), we subset our 16S rDNA sequences to a minimum sequence count of 3,277 to retain the maximum number of samples. However, when comparing only sediment samples, the 16S rDNA sequences were randomly subset to 20,000 sequences using the single_rarefaction.py QIIME script.

## Data visualization and statistical analyses

Data visualization was performed exclusively in R and statistical analyses were performed using a combination of QIIME scripts and R (*R Core Team, 2016*). For analysis done in R, the rarefied OTU tables were converted to json format and exported for analysis using the ggplot2 (*Wickham, 2009*), vegan (*Dixon, 2003*) and phyloseq (*McMurdie & Holmes, 2013*) packages. Initial analysis indicated no significant differences between the microbial communities associated with shallow (0.5 cm or less) and deep (3 cm) sediment samples, thus sediment depth was not considered further here. We describe the different types of analyses below.

- Intra-sample (alpha) diversity. We were interested in if significant differences existed between the intra-sample (alpha) diversities (richness, evenness) of the microbial communities associated with different sample types (leaf, root, sediment) and different sediment locations (inside, edge, outside). We calculated the following diversity metrics: Chao1 (*Chao, 1984*), Observed OTUs, Shannon (*Shannon & Weaver, 1949*) and Simpson Indices (*Simpson, 1949*) in R. To determine if there were significant differences between the alpha diversities of different sample types and different sediment locations, we first performed Kruskal–Wallis tests. We then implemented Bonferroni corrected post-hoc Dunn tests to identify which pairwise comparisons were driving differences.
- Inter-sample (beta) diversity. We assessed the inter-sample (beta) diversities of the microbial communities associated with different sample groupings (sample type, location, etc) and if there were any significant correlations between environmental variables and community dissimilarity. We used both Unifrac (weighted and unweighted) (*Lozupone et al., 2007*; *Hamady, Lozupone & Knight, 2010*) and Bray–Curtis (*Bray*

& *Curtis, 1957*) dissimilarities calculated in R using phyloseq. These dissimilarities were then plotted using principal coordinate analysis (PCoA) and non-metric multidimensional scaling (NMDS) methods. Multiple tests were then performed on these beta-diversity results. To test for significant differences in centroids between different sample groupings (sample type, location, etc.) PERMANOVA tests were performed using the adonis function from the vegan package in R with 9,999 permutations (*Anderson, 2001*). PERMANOVA tests can be sensitive to differences in dispersion when using abundance-based distance matrices (*Warton, Wright & Wang, 2012*), but are more robust than other tests, especially for balanced designs (*Anderson & Walsh, 2013*). To test for differences in mean dispersions between different groupings, the betadisper and permutest functions from the vegan package in R were used with 999 permutations. To test for correlations between the Bray Curtis dissimilarities of our samples and the environmental factors (C:N ratio, pH, etc) measured, euclidean distances were calculated in R using vegan and Mantel tests were performed using 9,999 permutations. The supervised_learning.py QIIME script was used to see if a random forest classifier could differentiate between sample type or sediment location using leave-one-out cross validation and 1,000 trees.

- Taxonomic variation. To determine if the mean relative abundance of taxonomic orders varied significantly between different sample types and sediment locations, we first used the summarize_taxa.py QIIME script to remove rare OTUs (less than one percent of total abundance) and to collapse OTUs at the Order level. We then used the group_significance.py QIIME script on the resulting OTU table to test for differences using Bonferroni corrected Kruskal–Wallis tests with 1000 permutations. We removed the rare OTUs, as suggested in the documentation for the groups_significance.py QIIME script, to avoid spurious significance from very low abundance OTUS, to simplify analyses and to focus on abundant organisms and overall patterns.

- Environmental variation. To determine if environmental factors varied significantly between different locations in the eelgrass patch (inside, edge, outside), ANOVA tests were performed in R for each factor. The post-hoc Tukey's Honest Significant Difference (HSD) test was performed in R for factors found significantly different by the ANOVA (*Tukey, 1953*; *Kramer, 1956*; *Kramer, 1957*).

## RESULTS & DISCUSSION

### Diversity metrics I: intra-sample variation between sample types and locations

Alpha diversity is greater in the sediment than in the leaves and roots ($p < 0.001$) for a variety of metrics including observed number of OTUs, Chao1, Shannon and Simpson (Fig. 1). However, there is no difference in alpha diversity between leaf and root samples ($p > 0.05$) (Table S1 ). This is not altogether unexpected as in terrestrial systems soil has been observed to have increased diversity compared to host associated sample types (*Edwards et al., 2015*). There is conflict between the diversity metrics when determining if the intra-sample diversity of sediment at different locations (inside, edge, outside) varies

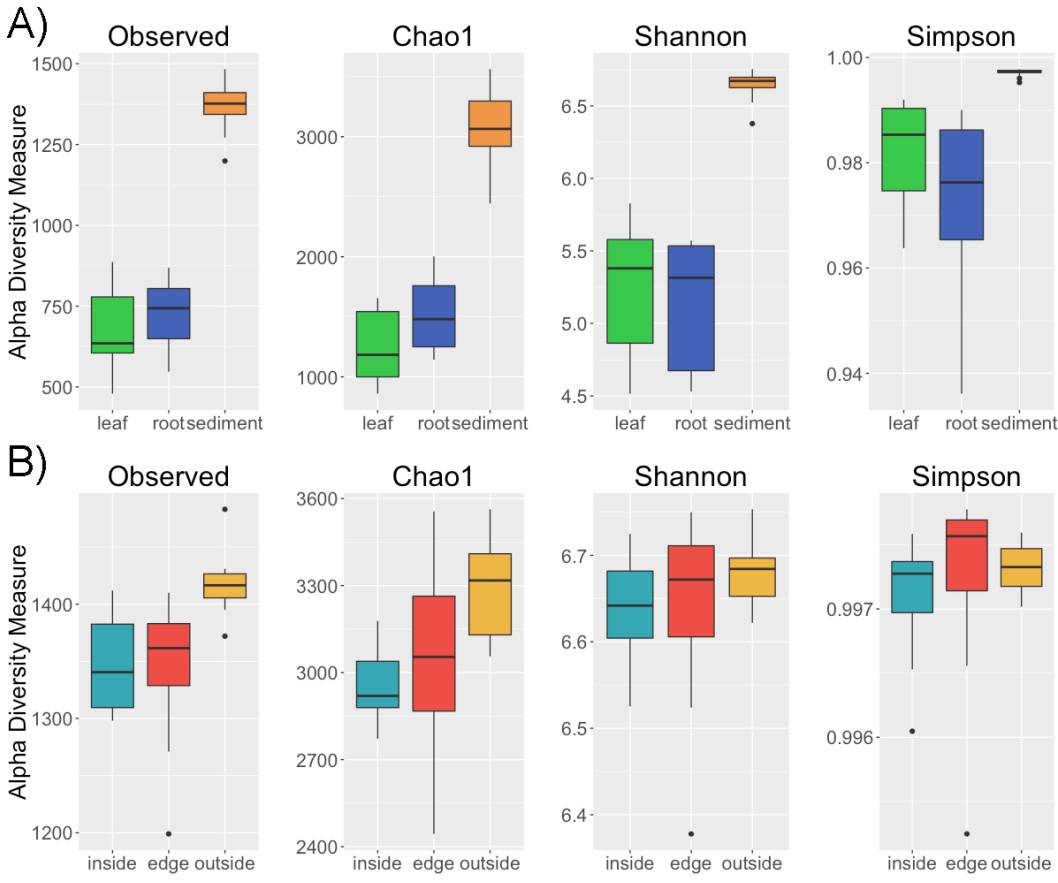

**Figure 1 Alpha diversity across samples.** Four alpha diversity metrics, observed number of OTUs, Chao1, the Shannon and Simpson diversity indices, are shown as boxplots for (A) different sample types (leaf, root, sediment) and for (B) sediment from different locations (inside, edge, outside).

(Table S2). Two of the metrics, observed number of OTUs and Chao1, indicate greater diversity outside compared to inside the patch ($p < 0.05$). The non-significant metrics, the Shannon and Simpson indices, account for both richness and evenness and are less sensitive to rare taxa than richness only metrics (*Bent & Forney, 2008*). Thus, one possible explanation for the difference in diversity between the inside and outside sediment is an increased number of rare taxa in sediment from outside the patch. No significant differences were found between the alpha diversity of leaves and roots between the inside and edge of the eelgrass patch.

## Diversity metrics II: inter-sample variation between sample types and locations

Distinct microbial communities were detected in association with *Z. marina* leaves, roots and sediment (Fig. 2). PERMANOVA tests performed on three different beta diversity metrics, weighted UniFrac, unweighted UniFrac and Bray–Curtis Dissimilarity, found these communities to be significantly different from each other with $p = 0.0001$ (Table 1). Root and leaf associated communities were found to have more with-in group variance, or dispersion, than sediment communities ($p = 0.001$), which could indicate that stabilizing

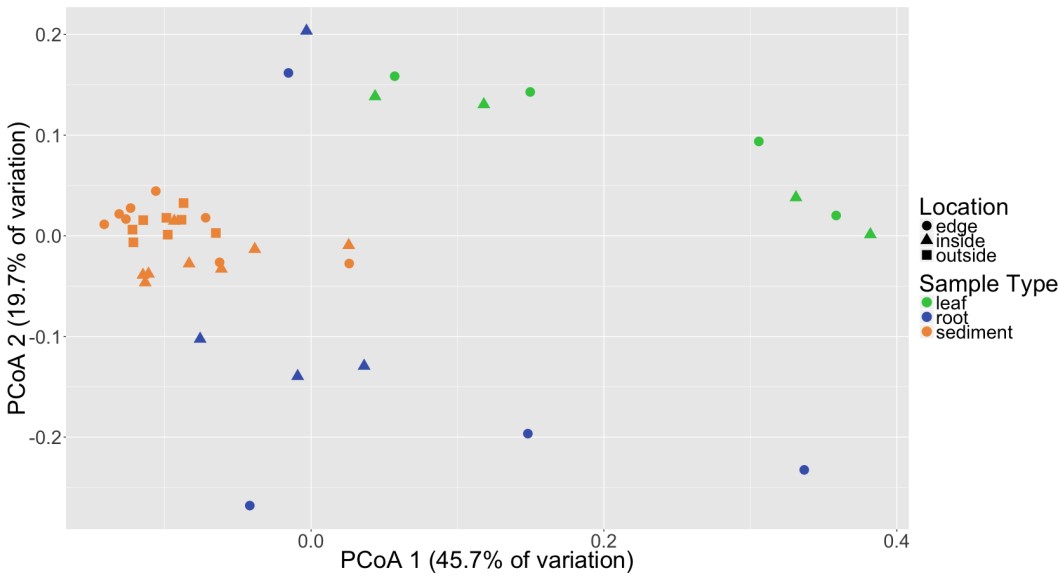

**Figure 2 Principal coordinates analysis (PCoA) of microbial communities based on weighted Unifrac distances.** Samples are colored by sample type (leaf, root, sediment) with different shapes for location (inside, edge, outside).

**Table 1 Permanova results.** Comparing microbial community composition between different sample types (leaf, root, sediment) and locations (inside, edge, outside) using multiple beta diversity metrics.

| Category | Diversity metric | Pseudo-F | R2 | P (perm) |
|---|---|---|---|---|
| Location | Weighted UniFrac | 2.22 | 0.107 | 0.0213 |
| | Unweighted UniFrac | 1.91 | 0.0938 | 0.0043 |
| | Bray Curtis | 2.82 | 0.133 | 0.0009 |
| Sample type | Weighted UniFrac | 13.75 | 0.426 | 0.0001 |
| | Unweighted UniFrac | 6.16 | 0.249 | 0.0001 |
| | Bray Curtis | 9.53 | 0.34 | 0.0001 |
| LocXType | Weighted UniFrac | 1.98 | 0.0541 | 0.0426 |
| | Unweighted UniFrac | 1.19 | 0.0455 | 0.1586 |
| | Bray Curtis | 1.482 | 0.0458 | 0.0795 |

selection is acting on these sediment communities. Random forest analysis further validated the observed differences between leaves, roots and sediment microbial communities (Table S3). The classifier had an estimated error of 5% (versus a baseline error of 40%) and correctly identified all leaf samples ($n = 8$) and all sediment samples ($n = 24$). The classifier did misclassify two of the root samples ($n = 8$) as leaves, but this is not unexpected as these two samples also appear to cluster more closely with the leaf samples when visualized using Principal Coordinates Analysis (PCoA) (Fig. 2). The reason that these root samples cluster more closely with the leaf samples may be due to which root bundles were sampled; preliminary results indicate that the microbiota associated with the roots can vary depending on the proximity of the root to the base of the leaf, with roots closer to the base appearing more "leaf-like" (HE Holland-Moritz et al., 2017, unpublished data).

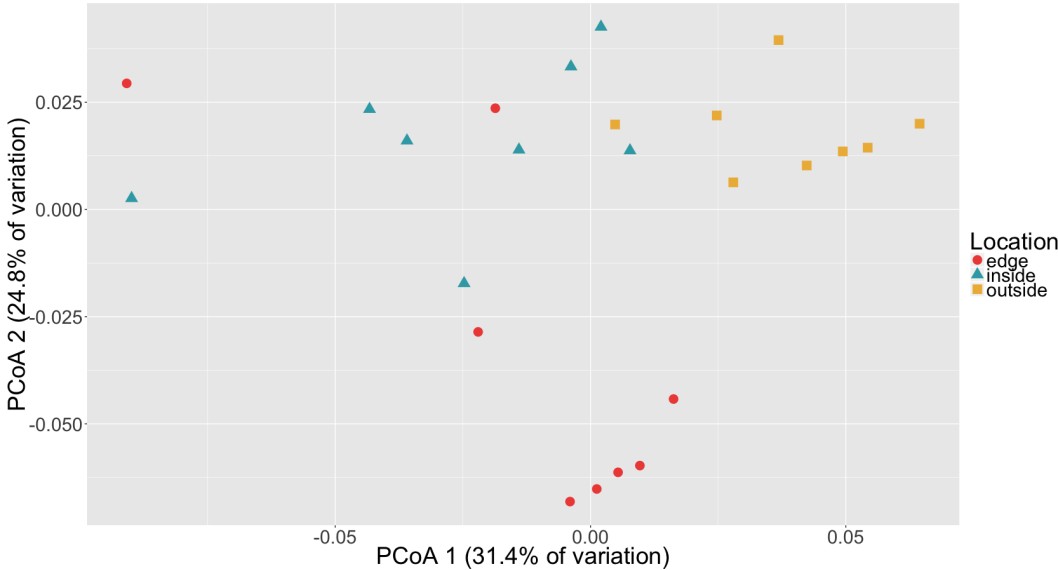

**Figure 3** **Principal coordinates analysis (PCoA) of microbial communities in sediment based on weighted Unifrac distances.** Samples are colored by location (inside, edge, outside).

**Table 2** **Sediment PERMANOVA results.** Comparing sediment microbial community composition between different locations (inside, edge, outside) and eelgrass densities using multiple beta diversity metrics.

| Category | Diversity metric | Pseudo-F | R2 | P (perm) |
|---|---|---|---|---|
| Location | Weighted UniFrac | 8.69 | 0.453 | 0.0001 |
| | Unweighted UniFrac | 2.92 | 0.217 | 0.0001 |
| | Bray Curtis | 8.01 | 0.433 | 0.0001 |
| Density | Weighted UniFrac | 2.81 | 0.551 | 0.0002 |
| | Unweighted UniFrac | 1.51 | 0.398 | 0.0001 |
| | Bray Curtis | 2.86 | 0.555 | 0.0001 |

To determine if there was a difference in community composition at the eelgrass patch edge relative to the inside or outside of the patch, beta diversity metrics were calculated for the sediment microbial communities. As can be seen in Fig. 3, these diversity metrics show the communities clustering by sampling location (inside, edge, outside). PERMANOVA tests indicate that these clusters are significantly different between locations ($p = 0.0001$) and also for eelgrass shoot densities ($p < 0.0002$) (Table 2). However, leaf and root microbial communities do not differ significantly based on sampling location, possibly indicating that these plant tissue associated communities are more stable than the sediment communities in regards to location. Whereas sediment communities, although distinct when associated with eelgrass, may be under less selection from the host plant. One possible explanation for the correlation between the sediment communities and eelgrass shoot density may be the release of exudates and oxygen by the roots of the eelgrass, which would increase in concentration with eelgrass density.

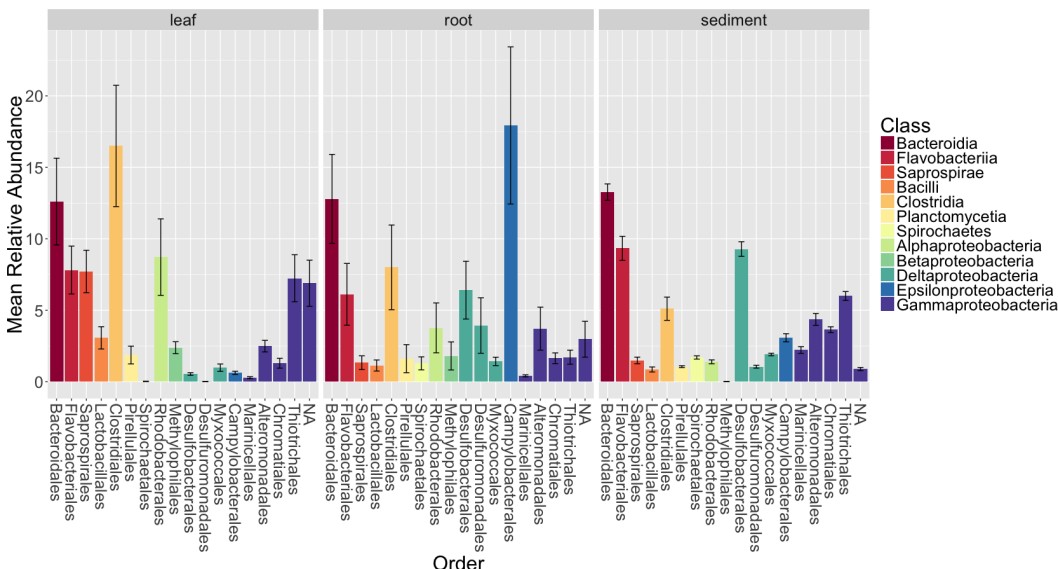

**Figure 4** **Average relative abundance of taxonomic groups associated with each sample type (leaf, root, sediment).** OTUs are shown grouped by taxonomic order and colored by taxonomic class. Only orders with a mean abundance of at least one percent are shown here. The bars represent the standard error of the mean.

Random forest analysis confirmed differences between sediment microbial communities taken from the inside of the patch, the edge and unvegetated sediment from outside the patch (Table S4). The classifier had an estimated error of 12.5% (versus a baseline error of 66.7%) and correctly identified all of the unvegetated sediment ($n = 8$). The classifier did mistakenly classify one sample from the edge ($n = 8$) as being from the inside of the patch and two samples from the inside of the patch ($n = 8$) as being from the edge. In Fig. 3, there is some overlap in the clustering of sediment from the inside and edges of patches which might account for these misclassifications.

## Major patterns in community composition of the leaves, roots and rhizosphere sediment

The analysis of diversity metrics presented above shows that there are distinct communities associated with leaves and roots, and these both differ from the sediment, whereas location effects are weaker. We therefore examined in more detail the taxonomic composition and possible functional roles of the microbes on *Z. marina* leaves, roots and rhizosphere sediment (sediment from the inside and edge of the eelgrass patch). We summarize our findings regarding this here.

Figure 4 shows the average relative abundance of different orders of bacteria for leaves, roots and sediment. On leaves, the most abundant orders were *Clostridiales*, *Bacteroidales*, *Rhodobacterales*, *Flavobacterales*, *Saprospirales*, *Thiotrichales* and Unidentified *Gammaproteobacteria*. On roots, the most abundant orders were *Campylobacterales*, *Bacteroidales*, *Clostridiales*, *Desulfobacterales*, *Flavobacteriales* and *Desulfuromonadales*. In the rhizosphere sediment, the most abundant orders were *Bacteroidales*, *Flavobacteriales*, *Desulfobacterales*, *Thiotrichales*, *Clostridiales* and *Alteromonadales*.
We also examined the overall patterns in our results at the class level (Table S5). For leaves, the most abundant class of epiphytes observed was *Gammaproteobacteria* (20.5 ± 7.3%). Other abundant classes included *Clostridia* (16.5 ± 12%), *Bacteriodia* (12.6 ± 8.6%), *Alphaproteobacteria* (11.4 ± 8.5%), *Flavobacteria* (7.8 ± 4.7%) and *Saprospirae* (7.7 ± 4.2%). For roots, the dominant class associated with the roots were *Epsilonproteobacteria* (17.9 ±15.5%). Other abundant classes observed on the roots include *Deltaproteobacteria* (13.4 ± 11.3%), *Bacteriodia* (12.8 ± 8.8%), *Gammaproteobacteria* (12.6 ± 11%), *Clostridia* (8 ± 8.4%), *Flavobacteriia* (6.1 ± 6.1%) and *Alphaproteobacteria* (4.8 ± 6.3%). In the rhizosphere sediment, the dominant class was *Gammaproteobacteria* (18.2 ± 3.4%), as it was on the leaves. Other abundant classes found in the rhizosphere sediment include *Deltaproteobacteria* (14.9 ± 2.6%), *Bacteriodia* (13.3 ± 2.3%), *Flavobacteriia* (9.3 ± 3.4%), Clostridia (5.1 ± 3.3%) and *Anaerolineae* (3.9 ± 1.6%).

The summary results above allow a comparison to findings from a recent study on the rhizosphere sediment microbiomes of three seagrass species, including *Z. marina*, *Cúcio et al., 2016*. We chose to focus our comparison on the Cúcio et al. study because it is one of the more comprehensive culture independent studies of seagrasses. Overall, there are general similarities and differences when comparing the class-level patterns between the studies. The authors reported that the most abundant classes were *Gammaproteobacteria* (32–38% depending on the species sampled), *Deltaproteobacteria* (23–26%), and *Bacteroidia* (6–7%). These were the three most abundant classes in our sediment samples as well, but at different relative abundances (see above). These differences could be due to true differences in microbiomes in the sediments sampled, or due to the use of different primer sets, extraction methods, and sample collection strategies (among many other differences).

When examined at higher taxonomic ranks, the microbiome of the leaves of *Z. marina* shares some similarities with the microbiomes of various marine algae (e.g., kelp and seaweeds), with *Gammaproteobacteria* being the most abundant class in both cases (*Hollants et al., 2013*). However, these similarities are not seen at lower ranks (e.g., order, family, genus). This finding is similar to what has been observed between different marine algal microbiomes, with similarities observed at higher, but not lower taxonomic levels (*Hollants et al., 2013*; *Egan et al., 2013*). This is further supported by a recent study, which focused on surface-associated communities, that observed that the microbiomes of seagrass and seaweed species were host specific, but had broad-scale functional similarities (*Roth-Schulze et al., 2016*).

### Differences in microbial communities between sample types (leaves, roots and rhizosphere sediment) and possible functional implications

We used a Bonferroni corrected Kruskall–Wallis test to test for differences in relative abundance of the orders between sample types. This showed that *Saprospirales*, *Thiotrichales*, *Rhodobacterales*, *Desulfobacterales*, *Desulfuromonadales*, *Marinicellales*, *Spirochaetales*, *Chromatiales* and *Campylobacterales* are significantly different between sample types ($p < 0.05$). *Campylobacterales*, *Desulfobacterales*, *Spirochaetales* and *Desulfuromonadales* were enriched on *Z. marina* roots. *Thiotrichales*, *Rhodobacterales* and *Saprospirales* were enriched on the leaves. *Thiotrichales*, *Marinicellales*, *Chromatiales*, *Desulfobacterales* and

*Spirochaetales* were enriched in the rhizosphere. We note that many of the taxa that differ significantly between communities are closely related to ones commonly associated with various aspects of sulfur and nitrogen metabolism. This is interesting because prior studies have suggested that nitrogen and sulfur metabolism are critical functions for the seagrass associated microbiome (*Lovell, 2002*). For example, acquisition of nitrogen (in its many forms) is frequently a limiting factor for the health of plants, including seagrasses (*Short, 1987*; *Elser et al., 2007*) and associations with microbes are frequently critical for such acquisition (*Welsh, 2000*; *Nielsen et al., 2001*). In addition, since the reduced sulfur compounds that accumulate in aquatic sediments are known phytotoxins (*Lamers et al., 2013*), it is thought that sulfur metabolizing microbes could play important roles in aiding seagrass survival in such sediments (*Barber & Carlson, 1993*; *Terrados et al., 1999*; *Erskine & Koch, 2000*; *Van der Heide et al., 2012*). Sulfur and nitrogen metabolism are not necessarily independent—it has been postulated that sulfate-reducing bacteria may be responsible for most of the nitrogen fixation that occurs in seagrass sediments (*Capone, 1982*).

Given this context, we discuss several of the specific taxa that differ between samples and their possible connection to nitrogen and/or sulfur metabolism below. For example, *Campylobacterales*, specifically *Sulfurimonas* species, from the class *Epsilonproteobacteria*, were enriched on *Z. marina* roots. Previous studies of *Spartina* (*Thomas et al., 2014*) and *Z. marina* (*Jensen, Kühl & Priemé, 2007*) also found enrichment of *Epsilonproteobacteria* on roots relative to the surrounding sediment. All known *Sulfurimonas* species are sulfur-oxidizing chemolithoautotrophs, can perform denitrification and are postulated to play significant roles in biogeochemical cycling in marine sediments (*Campbell et al., 2006*). Members of *Campylobacterales* have previously been identified as nitrogen fixers when isolated from *Spartina* roots (*McClung & Patriquin, 1980*). Additionally, *Campylobacterales* and *Desulfobacterales*, known sulfur-reducing bacteria, have been previously found to be abundant in association with plants from brackish habitats (e.g., mangroves—*Gomes et al., 2010*). *Rhodobacterales* which are enriched on the *Z. marina* leaves in our study, are purple nonsulfur bacteria, that have been identified as primary surface colonizers in marine habitats and have been shown to have the ability to fix nitrogen (*Palacios & Newton, 2005*; *Dang et al., 2008*). *Desulfobacterales* and *Rhodobacterales* species have been previously found in association with the tropical seagrass *Thalassia hemprichii* (*Jiang et al., 2015*). *Thiotrichales*, which are enriched in the sediment, are generally filamentous sulfur-oxidizing bacteria (*Garrity, Bell & Lilburn, 2005*) and are postulated to be dominant sulfur-oxidizers in salt marsh sediments (*Thomas et al., 2014*).

## Variation in sediment microbial communities between locations

The analysis of diversity metrics reported above also showed that there are significant differences in the sediment microbial communities from different locations (inside a eelgrass patch, the edge of a patch and outside of a patch). We therefore examined in more detail the taxonomic groups that differ significantly between sediment locations and their potential functional roles (Fig. 5).

*Bacteroidales*, *Myxococcales*, *Thiotrichales* and *Chromatiales* are significantly different between locations with a Bonferroni corrected Kruskall–Wallis test ($p < 0.01$). *Thiotrichales*
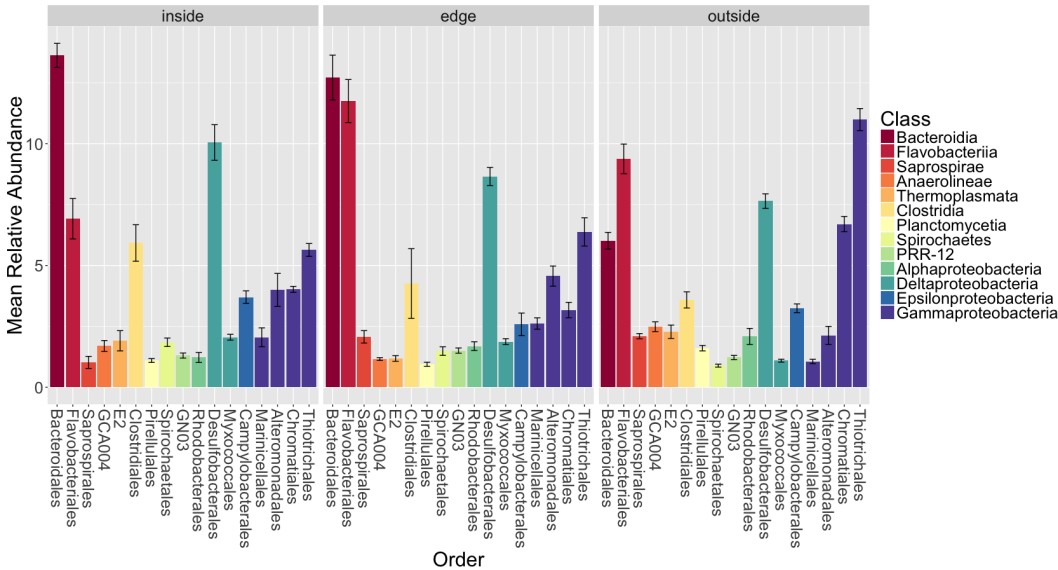

**Figure 5** **Average relative abundance of taxonomic groups associated with sediment from each location (inside, edge, outside).** Operational taxonomic units (OTUs) are shown grouped by taxonomic order and are colored by taxonomic class. Only orders with a mean relative abundance of at least one percent are shown. Bars represent the standard error of the mean.

and *Chromatiales* are enriched outside of *Z. marina* patches in the unvegetated sediment compared to the inside or edge of patches. In contrast, *Bacteroidales* and *Myxococcales* are enriched in the rhizosphere sediment inside and at the edge of eelgrass patches compared to the outside. The functional significance of these differences is unclear but we note a few things here. First, *Thiotrichales* and *Chromatiales* are common taxa in other marine and brackish sediments including those associated with various plants (e.g., *Thomas et al., 2014*). This is thought be reflective of a role in sulfur oxidation (see discussion above). Some studies have indicated that these taxa are associated with plants (e.g., seagrasses in Portugal *Cúcio et al., 2016*). However, other studies have indicated that these are found more in the sediment near plants but not specifically associated with plants (*Thomas et al., 2014*). *Myxococcales,* commonly found in freshwater and marsh sediments, includes microorganisms known to be involved in organic matter degradation (*Bowen et al., 2012*; *Kou et al., 2016*; *Cleary et al., 2016*). The abundance of *Myxococcales* inside the eelgrass patch aligns with the expectation of higher prevalence of organic matter degradation inside the patch as opposed to surrounding unvegetated sediment.

## Environmental drivers of sediment communities

In addition to investigating the taxonomic composition of the microbial communities of sediment collected from the inside, edge and outside of eelgrass patches, we decided to test for correlations between observed community differences and environmental factors to elucidate key factors that may be driving the microbial communities in eelgrass patches.

A variety of abiotic factors were significantly different between locations including C:N ratio, TIC, dissolved oxygen, pH and sediment size fractions 710 μm and 63 μm (ANOVA, $p < 0.05$) (Tables S6 and S7). Unsurprisingly, eelgrass shoot density was significantly

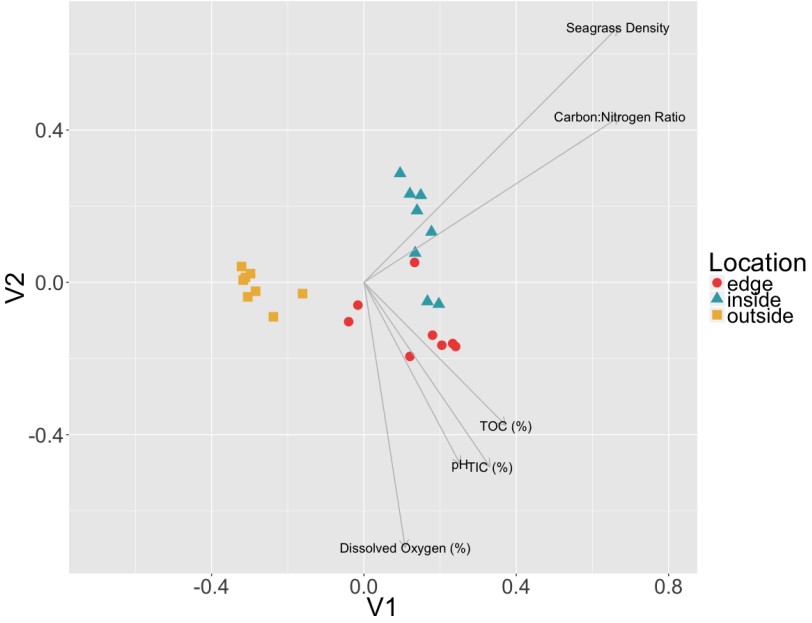

**Figure 6  Relationship between environmental data and microbial communities.** Non-metric multi-dimensional scaling (NMDS) of Bray Curtis dissimilarities of microbial communities found in sediment samples are shown here colored by location (inside, edge, outside). Environmental factors ($p < 0.055$, ANOVA) were overlaid as vectors onto the NMDS using the envfit function in vegan.

different between locations (ANOVA, $p < 0.05$). To determine which pair-wise locations were driving the significant differences between location overall, we performed Tukey's HSD tests (Tables S8 and S9). We also performed Tukey's HSD tests on percent TOC which was marginally non significantly different (ANOVA, $p = 0.0519$). All pair-wise location comparisons of eelgrass density and dissolved oxygen were significantly different (Tukey's HSD, $p < 0.05$). The C:N ratio and sediment fraction 63 μm were significantly different for the outside-inside comparison (Tukey's HSD, $p < 0.05$). Percent TIC and TOC as well as sediment fraction 710 μm were significantly different for the outside-edge comparison and pH was significantly different for the inside-edge comparisons (Tukey's HSD, $p < 0.05$).

To test if there was a correlation between environmental measures and microbial community composition, Mantel tests were performed on Euclidean distances of environmental measures and the Bray–Curtis dissimilarities of sediment communities. A combined dataset including C:N ratio, TIC, TOC, dissolved oxygen, pH and eelgrass density was found to be significantly positively correlated with the sediment microbial community data ($r = 0.1122$, $p = 0.0474$) (Fig. 6). However, when measures were tested individually only the C:N ratio ($r = 0.1701$, $p = 0.016$) and eelgrass density ($r = 0.1292$, $p = 0.0381$) were significantly correlated with microbial community composition.

The significant correlation between the sediment communities and the C:N ratio may indicate a change in ecosystem nutrient cycling at the patch edge. As Carbon (TIC and TOC) was not correlated with the sediment microbial communities, the correlation with the C:N ratio may hint at the importance of nitrogen, which was not measured in this study, to sediment community composition. Nitrogen is often a limiting terrestrial plant nutrient

and N-limitation has also been observed in several seagrass studies, more frequently in temperate habitats (*De Boer, 2007*). Terrestrial plants overcome N-limitation by having beneficial interactions with nitrogen fixing bacteria and these bacteria have previously been observed to form associations with eelgrasses (*Capone & Budin, 1982*; *Welsh, 2000*; *Bagwell et al., 2002*; *Adhitya, Thomas & Ward, 2007*; *Sun et al., 2015*). Nitrogen fixation can account for 5–10% of plant nitrogen requirements in temperate eelgrass patch sediments, and up to 50% in tropical patch sediments, indicating an important role for nitrogen fixation in overall patch health (*Welsh, 2000*).

A previous study looking at forest soil microbial communities found that microbial biomass and activity were significantly lower at forest edges due to decreased litter decomposition in the edge habitat and thus, changes in nutrient cycling (*Malmivaara-Lämsä et al., 2008*). In seagrass patches, on average, vegetated sediments are significantly enriched in organic matter compared to unvegetated sediments, with carbon stocks generally higher on the inside of patches (*Duarte, Holmer & Marbà, 2005*; *Ricart et al., 2015*). It is possible that the observed community structure changes in the sediment between locations and the correlations with C:N ratio and eelgrass density here are indicative of a similar trend of location based nutrient cycling resulting from differing nutrient deposition and decomposition rates.

Eelgrass density may have direct or indirect effects on sediment microbial communities as a result of the role eelgrass plays in its environment as a foundation species and an ecosystem engineer (*Koch, 2001*) Seagrasses are known to modify their surrounding habitat in a variety of ways including enhancing the input and retention of carbon and other nutrients in the sediment (*Gacia et al., 2002*; *Duarte et al., 2005*; *Duarte & Cebrián, 1996*), altering flow velocity and turbulence in the water column above patches (*Fonseca et al., 1982*; *Granata et al., 2001*; *Folkard, 2005*) which can increase sedimentation (*Short & Short, 1984*; *Dauby et al., 1995*; *Gacia et al., 2002*) and oxygenating the sediment using their roots (*Caffrey & Kemp, 1991*; *Pedersen et al., 1998*; *Connell, Colmer & Walker, 1999*).

Other factors at play in the observed differences between locations as a result of eelgrass density may be the development stage of the eelgrass at the edge (if the patch is expanding or contracting) and the variable use of eelgrass as a habitat by macroorganisms. From terrestrial systems, it is known that microbial communities can vary across different stages of plant development (*Chaparro, Badri & Vivanco, 2014*). Seagrasses at earlier stages in development are known to have different carbon deposition rates than later stages, so if seagrass patch was in the process of expanding this may change the abiotic conditions at the patch edge, and thus might be reflected in distinct microbial communities at the edge of a patch compared to the inside (*Duarte & Sand-Jensen, 1990*; *Cebrián et al., 2000*). Additionally, seagrass patches are habitats for a large number of macroorganisms with variable abundance across seagrass patch landscapes (seagrass densities) (*Tanner, 2005*; *Smith et al., 2008*; *Smith et al., 2011*).

Ultimately, although we see differences between locations in environmental abiotic measurements, we are unable, given the limitations of this study, to decouple these measurements from the eelgrass itself (eelgrass density), which is highly correlated with sediment community composition.

## CONCLUSIONS

This study provides new insights into the composition and assembly of the *Z. marina* microbiome. Major findings include that distinct microbial communities are associated with the leaves and roots of the plant, sediment associated communities are correlated with host plant density,and specific microbial taxa are found to have high relative abundances on particular tissues. Differences in the rhizosphere sediment community composition at the patch edge were observed and correlated with variation in environmental measurements. However, we were unable to disentangle these measures from eelgrass density, with the strongest correlated factor with community differences being presence/absence of the host plant. This is perhaps not unexpected from a field study, as eelgrass species are ecosystem engineers that actively change the sediment chemistry and landscape (*Orth et al., 2006*; *Bos et al., 2007*).

Overall, we believe that the results of this study hint at a network of complex interactions between *Z. marina*, the microbes associated with *Z. marina* and biogeochemical cycling. Untangling such networks is difficult but increasingly feasible. Although *Z. marina* is not a model organism in the sense of *Arabidopsis* or poplar, we believe it can nevertheless become a model for host-microbiome-environment interaction studies. Advantages of working on this species include that there is a genome now available (*Olsen et al., 2016*), that there is a large network of collaborating labs focusing on this species (Zostera Experimental Network; http://zenscience.org), and that it can be used in common garden and reciprocal transplant experiments. Along these lines we have been building a library of cultured isolates associated with this species and sequencing the genomes of many of these (*Lee et al., 2015a*; *Lee et al., 2015b*; *Lee et al., 2016a*; *Lee et al., 2016b*; *Alexiev et al., 2016a*; *Alexiev et al., 2016b*). There are still areas in need of improvement (e.g., there a limited amount of full length 16S and 18S other reference data; only limited information on the in situ functions of microbes are available, there is a need for more genetic tools for the host), but given the importance of coastal marine systems and seagrasses generally, we believe continued efforts to study the host-microbiome-environment interactions in this and related species are important.

## ACKNOWLEDGEMENTS

Illumina sequencing was performed at the DNA Technologies Core facility in the Genome Center at UC Davis, Davis, California. We thank Qingyi "John" Zhang for his help with the Illumina library preparation and Hannah Holland-Moritz for her help with DNA extractions. We thank Ted Grosholz and Susan Williams for hosting Sofie Voerman during her visit and for help with the sampling design. We thank Daniel Bradley for his help with sample collection.

### Funding

This work was supported by a grant from the Gordon and Betty Moore Foundation (GBMF333) "Investigating the co-evolutionary relationships between seagrasses and their

microbial symbionts.'' The funders had no role in study design, data collection and analysis, decision to publish, or preparation of the manuscript.

### Grant Disclosures
The following grant information was disclosed by the authors:
Gordon and Betty Moore Foundation: GBMF333.

### Competing Interests
Jonathan A. Eisen is an Academic Editor for PeerJ. Jenna M. Lang is an employee of Trace Genomics, Inc.

### Author Contributions
- Cassandra L. Ettinger analyzed the data, wrote the paper, prepared figures and/or tables, reviewed drafts of the paper.
- Sofie E. Voerman conceived and designed the experiments, reviewed drafts of the paper, performed sampling.
- Jenna M. Lang conceived and designed the experiments, reviewed drafts of the paper.
- John J. Stachowicz reviewed drafts of the paper, advised on experimental design, edited drafts of paper.
- Jonathan A. Eisen contributed reagents/materials/analysis tools, reviewed drafts of the paper, advised on data analysis, edited drafts of paper.

### DNA Deposition
The following information was supplied regarding the deposition of DNA sequences:
This 16S rRNA sequencing project has been deposited at GenBank under the accession no. PRJNA350006.

### Data Availability
Coil, David; Eisen, Jonathan; Stachowicz, Jay; Green, Jessica; Holland-Moritz, Hannah; Lang, Jenna (2014): The Seagrass Microbiome. figshare.
https://doi.org/10.6084/m9.figshare.1014334.v1.

### Supplemental Information
Supplemental information for this article can be found online at http://dx.doi.org/10.7717/peerj.3246#supplemental-information.

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
