# Peer review of "Microbial communities in sediment from Zostera marina patches, but not the Z. marina leaf or root microbiomes, vary in relation to distance from patch edge"

_PeerJ, doi:10.7717/peerj.3246_

## Round 0.1 · original submission · Minor Revisions

The reviews are relatively favourable, but in particular, Reviewer 1 does raise a number of issues that need to be addressed appropriately. Please respond to all the comments in your revision.

·

Basic reporting

The manuscript entitled "The Zostera marina rhizosphere microbiome, but not the leaf or root associated communities, varies in relation to distance from edge" by Ettinger et al. describes an analysis of root, shoot, and sediment microbiomes associated with patches of eelgrass, where they distinguish those located more inside a patch, those at the edge, and for sediment also those outside of patches.

While the experiment is relatively modest (sampling 4 quadrats of eelgrass) the authors nicely show that there are some general differences between plants and sediments in terms of microbiome, and that within sediments the location in reference to the edge appears to make a difference.

The authors go on to argue that this may be a suitable model system to study marine plant-microbiome interactions, with the arguments that eelgrass already is a model system in many marine ecological studies, and because their results suggest that there may be some functional links in terms of nitrogen and sulfur cycling based on taxa found to associate with eelgrass. The latter is generally not overstated in the manuscript which it shouldn't, because we don't know whether 1) these organisms are on or in the eelgrass, 2) whether they do perform these functions, and 3) whether it has any effect on plant performance/fitness. As the authors acknowledge sampling one seagrass-bed, and not contrasted against other plants, is not enough to really conclude anything about host or tissue specificity.

Concerning the statistics and interpretation, this is generally appropriate and well described. However, judging from Fig 2 there appears to be quite a difference in multivariate dispersion between sediment- and plant associated communities. As Warton et al. (2012) report PERMANOVA may be sensitive to this, so I suggest to consider this in your interpretation. Potentially this higher variation among communities can be discussed as well?

Concerning the title: currently sediment microbial communities are reffered to as "rhizosphere", but in the manuscript they are generally referred to as sediment. The latter is more correct in my opinion, as no effort was taken to necesserily sample those associated with roots.

Concerning language, in quite some instances an additional round of proofreading would be good. Currently there are many small type-o's and awkwardly structured sentences. I put some examples below under comments and suggestions.

Concluding, after incorporating these suggestions I think the MS would make a nice addition to the literature and can serve as a starting point to look deeper into the marine angiosperm microbiome.

Experimental design

no comment.

Validity of the findings

no comment

Additional comments

Specific comments and suggestions

L64: here it claimed that because there are 64 seagrass species this makes them an interesting evolutionary study subject. Why is that? Because the low number makes it more tractable, because one could achieve full coverage?

L107: It is clear that the authors consider bacteria to constitute the microbiome (as opposed to also including archeae, fungi, protists). This is fine, but it would be good to state that here clearly and acknowledge there are other taxa that closely interact with plants.

L131: please rephrase, "simultaneously when" into e.g. "microbiome and samples and environmental data were collected simultaneously".

L133: YSI meter; manufacturer?

L140: TIC=TOC

L143: how were leaves and roots handled during DNA isolation? Were they just put in the tube fresh from the freezer (as the writing suggests) or first e.g. ground using liquid N? Need some more info here.

L182: Previously it is mentioned that the sample with the lowest read number apart from the control had about 23,000 reads. Were so many lost subsequently that rarefying to approx. 4000 is necessary?

L273: " However, leaf and root microbial communities do not differ significantly based on sampling location, possibly indicating that these plant tissue associated communities are more stable or are more advantageous for the host plant than the sediment communities." – why should the observation that leaf and root associated communities don't vary according to location indicate that they are advantageous, or that they are more advantageous than sediment communities?

L278: "which would be scale" ?????

L295: "epiphyte" here refers to the fact that they are found on shoots/roots, right? That makes it redundant, and potentially confusing given that it implies an independent denomination of the taxa.

L398: "Unsurprisingly, eelgrass shoot density was significant between locations (ANOVA, p < 0.05)" – this should be "significantly different". See for more instances in the following sentences.

L416: " The significance of the Carbon:Nitrogen ratio may be indicative of a change in ecosystem nutrient cycling at the bed edge and the importance of nitrogen to the system as Carbon (TIC and TOC) alone was not correlated with the sediment communities." – but I presume neither was nitrogen? Here follows a paragraph on the importance of certain microbes for plant nitrogen nutrition, which is interesting, but only very passingly relates to the observed significant relationship of C:N ratio with microbial community turnover.

L435: what's a "detritus rate"?

L465: "but distance from edge" – something is missing here.

References:
Please double check references. I can see that e.g. for Aleklett et al. The first and family names of some authors have been inverted (e.g. Noah F, Miranda H =Fierer and Hart).

refs cited above
Warton DI, Wright ST, Wang Y. 2012. Distance-based multivariate analyses confound location and dispersion effects. Methods in Ecology and Evolution 3: 89–101.

Reviewer 2 ·

Basic reporting

The authors analyzed the Zostera microbiome by means of 16S rRNA sequencing (Myseq). The aim of the study was to determine the epiphytic and rhizospheric microbial community composition. Differences were disclosed after comprehensive statistical analyzes and related to possible ecologic roles played by Zostera microbes. Overall, it is an excellent study, however it lacks a reference/discussion to previous relevant work on seawaeed (e.g. in Delisea). Eg PMID 22775757, 23157386, 22985125, 25675000.

Experimental design

ok, but i did not find 16S analyzes of the surrounding seawater of at least use of previously obtained/published 16S profiles from water column.

Validity of the findings

.

Additional comments

.

---

## Round 0.2 · accepted · Accept

The reviewers are happy with the revisions you made, and therefore the manuscript is acceptable for publication in PeerJ.

·

Basic reporting

I have re-read the manuscript and I carefully went through the rebuttal and checked which changes were made. I believe the authors have done an excellent job in improving their manuscript and think it now makes a great contribution to the field. I have no further suggestions to improve the work.

Experimental design

No comments

Validity of the findings

No comments